# Evaluation of protection benefit of sand barrier fence with different heights on desert highway

**Ming Zhang[1], Qi Li [2]\*, Jiayin Hou[3], Shuai Ji[2]**

**1** Gansu Engineering Design and Research Institute Co., Ltd, Lanzhou, China, **2** School of Civil Engineering, Lanzhou Jiaotong University, Lanzhou, China, **3** Gansu Road and Bridge Construction Group Co., Ltd, Lanzhou, China

\* liqi19991030@163.com

## Abstract

As the first barrier of desert highway protection, sand-blocking fence is very important to the safety of the line. Based on the background of Wuma Expressway, this paper uses CFD numerical simulation to study the wind and sand-blocking effect of sand-blocking fence with different heights. The results show that: (1) Between the first and second sand-blocking fences, when the height of sand-blocking fence is 2m, 2.5m, 3.0m and 3.5m, the wind speed near the surface (0.1m~0.3m) is reduced by 87%~97% of the initial wind speed. Between the second and third sand-blocking fences, when the height of sand-blocking fence is 2.5m, the increase of wind speed is 13.87% lower than that of 2m height. The decrease is the largest, and sand particles are easy to deposit here in large quantities. When the height is 2.5m and above, the windbreak efficiency is greater than 90%, and the windbreak effect is significantly improved. (2) The change of sand barrier height has a significant effect on the windbreak efficiency between the second and third sand barriers. (3) Among the three sand-blocking fences, when the height of the sand-blocking fence is 2.5m, the thickness of the sand is 50.51% and 58.33% higher than that of the 2m high sand-blocking fence, and the sand-blocking effect is the most significant. After the height is increased to 3.5m, the thickness of the sand is no longer increased. (4) The height of sand-blocking fence is 2.5m, and the area of sand at the top of embankment is obviously reduced. The area of sand volume fraction 0.55–0.6 is 73.44% lower than that of 2m sand-blocking fence, and the effect of wind and sand prevention is the best.

## 1 Introduction

The prevention and control of wind and sand disasters is a key issue that needs to be solved urgently in desert highways. As an important part of the protection system, sand barrier is an important barrier to block sand particles. Therefore, whether the parameter setting of sand barrier is reasonable or not is directly related to the

**Data availability statement:** All relevant data for this study are publicly available from the Zenodo repository (https://doi.org/10.5281/zenodo.15605881).

**Funding:** The author(s) received no specific funding for this work.

**Competing interests:** The authors have declared that no competing interests exist.

protective effect of the protection system and thus affects the safe operation of desert highways.

In recent years, great progress has been made in the study of wind and sand resistance benefits of sand barriers. Many scholars have studied the setting of sand barrier parameters by means of field observation, wind tunnel test and numerical simulation [1]. Computational Fluid Dynamics (CFD) modelling of wind flow has become a common tool to predict and understand secondary wind flow and resulting sediment transport [2]. Marko et al [3] evaluates the objective function by calculating wind engineering simulations. The sensitivity of barrier aerodynamics to design parameters was preliminarily evaluated. The performance based on gradient and genetic algorithm optimization is discussed. Yoshihide T et al [4] used computational fluid dynamics (CFD) analysis to provide validation data, and conducted wind tunnel experiments on sand erosion/ deposition around the cube. The results show that even when relatively simple sand transport modeling is performed, CFD can accurately reproduce the large-area erosion area near the windward angle of the cube.Based on the wind tunnel test, Li et al [5–7] measured the wind-sand flow before and after different types of sand-blocking sand barriers and sand-blocking walls, and evaluated their protection benefits through wind speed profile characteristics, wind speed flow field characteristics and sand transport rate. Luo et al [8,9] obtained the optimal configuration parameters of sand barrier by setting anemometers and sand traps between sand barriers with different configuration parameters, and calculating the effect of wind and sand barrier according to the measured data. An et al [10] conducted field observations on the erosion and deposition process of sand-blocking fences at different heights. The results showed that the upwind direction of sand-blocking fences at different heights showed wind-sand accumulation, and the downwind wind erosion and accumulation coexisted, and the degree was related to the height of the fence. The change trend of the overall erosion amount and erosion intensity of the fence was similar to that of the downwind direction. On the basis of wind tunnel experiments and field observations, Hu et al [11] carried out a systematic simulation study on the protection width of three different configurations of the protection system by using the system simulation method. The protection width of protection system with different configuration modes is obtained. Ling et al [12] carried out the experimental study on the sand blocking of the front fence of the protective belt on the north side of the railway in Shapotou area. The average sand blocking efficiency of the fence is 70–80%, and it is concluded that the sand blocking fence measures are feasible to block the sand accumulation in the front. Ding et al [13] analyzed the sand accumulation and sand-blocking efficiency of reed bar sand-blocking fence under different parameter settings by means of sand collector and three-dimensional laser scanner monitoring, and confirmed the correctness of 10~15H layout spacing of reed bar sand-blocking fence in Geku Railway. Kang et al [14] conducted a wind tunnel simulation test on the flow field structure of the sand-blocking fence in the Shapotou section of the Baotou-Lanzhou Railway. The results show that although porosity is an important technical parameter for the design of sand-blocking fence, the fence structure determines the key factor for the protection effect of wind-sand activities. Ma et al

[15] through anemometer and sand collector, if the high vertical sand barrier is raised, it can block the wind sand flow and form a larger scale of sand accumulation. The search for the optimal array of fences has remained largely an empirical task. In order to achieve maximal soil protection using the minimal amount of fence material, a quantitative understanding of the flow profile over the relief encompassing the area to be protected including all employed fences is required [16]. The findings have implication for a better understanding of aeolian transport in the presence of sand fences, as well as to develop optimization strategies for measures to protect soils from wind erosion [17]. In summary, scholars have confirmed the effectiveness of the protective effect of sand barrier, and used a variety of methods to analyze the protective benefits of sand barrier under different parameters, but there are few studies on the protective benefits and parameter optimization of sand barrier at different heights.

In view of this, based on the desert section project of Wuhai-Maqin (Wuma) expressway, this paper uses Fluent numerical simulation software to construct sand-blocking fence models with different heights, analyzes its wind-proof and sand-blocking effect, and then gives a reasonable height of sand-blocking fence. The research results can provide reference for mechanical sand control engineering.

## 2 Overview of the study area and research methods

### 2.1 Overview of the study area

The Wuma Expressway passes through the hinterland of the Tengger Desert from east to southwest. The desert section is about 18.1 km long. The starting point of the line is connected to the Qingtongxia section, and the end point is at the Hongwei hub. According to the law of wind and sand activity along the line, following the principle of adapting measures to local conditions, a sand control system with mechanical sand control measures as the guide and plant sand control measures as the core is proposed. The on-site protection system along the line is shown in Fig 1.

The protection system from the outside to the inside is: sand barriers, high vertical large mesh sand barriers, low vertical nylon mesh sand barriers, grass checkerboard sand barriers, gravel shelter belts. The sand barrier, as the outermost sand blocking measure of the protection system, is the first barrier to effectively block the wind and sand flow. Therefore, the reasonable layout of the sand barrier is related to the safe operation of the desert highway.

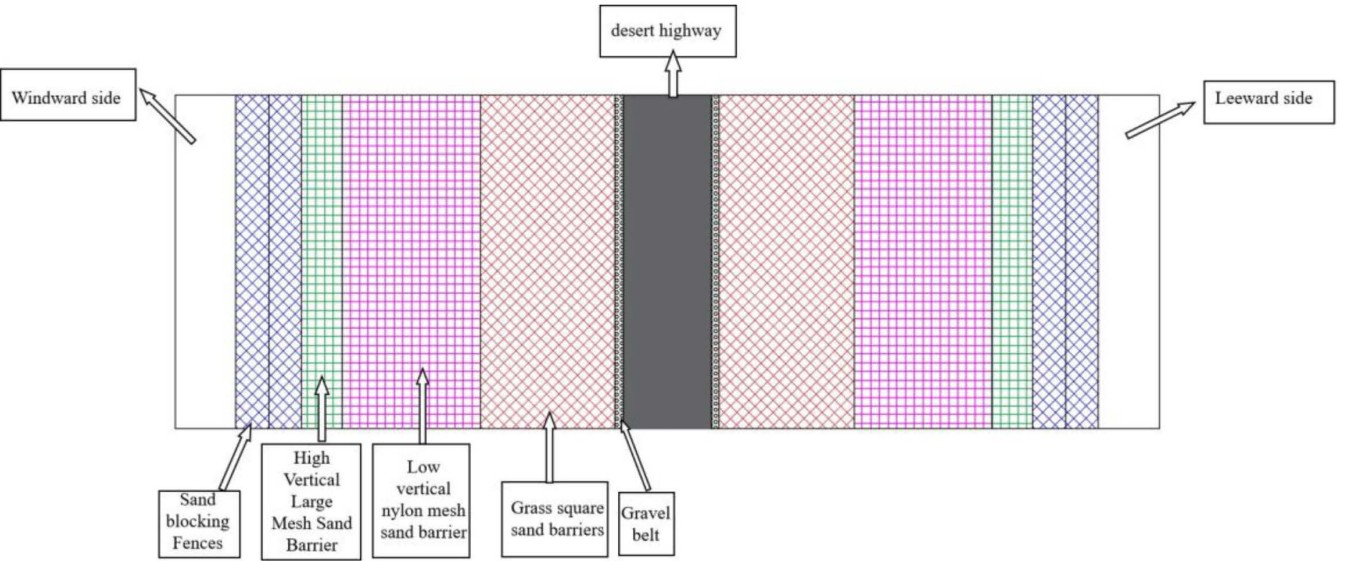

**Fig 1. Schematic sketch of the protection system of Wuma Expressway.**

## 2.2 Numerical simulation method

**2.2.1 Model construction.** As the first barrier of highway sand protection, it is of great significance to study the parameter improvement of sand barrier. In order to ensure that it is compatible with the field situation and reduce the amount of calculation, the geometric model includes a sand barrier and a high-rise large-grid sand barrier adjacent to the sand barrier. The main component of HDPE raw material is high-density polyethylene, which is processed by high-strength composite material combined with special weaving process. It has strong anti-ultraviolet radiation, frost resistance, decay resistance, acid, alkali, salt corrosion resistance, long service life and other characteristics. The unique scientific design structure of the sand-proof net can not only make the net body shake with the wind, but also make it have self-purification function and avoid dust accumulation.The HDPE sand barrier is simplified to a thin plate parallel to the X and Y directions according to the structural size. In the numerical simulation, it is set as a porous medium area, and its porosity is controlled by setting relevant parameters. In order to ensure the full development of the flow field, the rear end is connected to an 80-m-long fluid domain. The height of the sand barrier is 2m, and the newly calculated height values are: 2.5m, 3m, 3.5m, and the spacing is maintained at 25m. The porosity is still 60% [18,19] and the size of the fluid domain is: $X \times Y \times Z = 200m \times 24.15m \times 20m$. The model diagram is shown in Fig 2.

**2.2.2 Meshing.** In this paper, the protection system and embankment are meshed by ICEM and Fluent Mesh respectively. In order to ensure the quality of the mesh, the protection system is divided into blocks. Considering that the hexahedral mesh has high quality and small calculation error, the hexahedral mesh is used in the modeling process. The sand particles in the wind-sand flow are more in the near-surface height range and the distribution is more complex than the upper layer. At the same time, the boundary layer has a great influence on the wind-sand incoming flow. Therefore, the near-surface position of the model is encrypted during the meshing process, and the position near the top surface of the model is sparsely divided to save computing resources and speed up the operation. The total number of grids is about $1.58 \times 10^6$, and the orthogonal quality of the grid is about 0.9. The quality is good and meets the calculation requirements. The grid division is shown in Fig 3.

Since the sand-blocking measures such as sand-blocking fences are network structures, the porous media method is needed to characterize their porosity. The grid division is divided by array, and other division methods cannot accurately simulate porosity. Therefore, the grid independence verification is only for the subgrade. Three grid division methods are

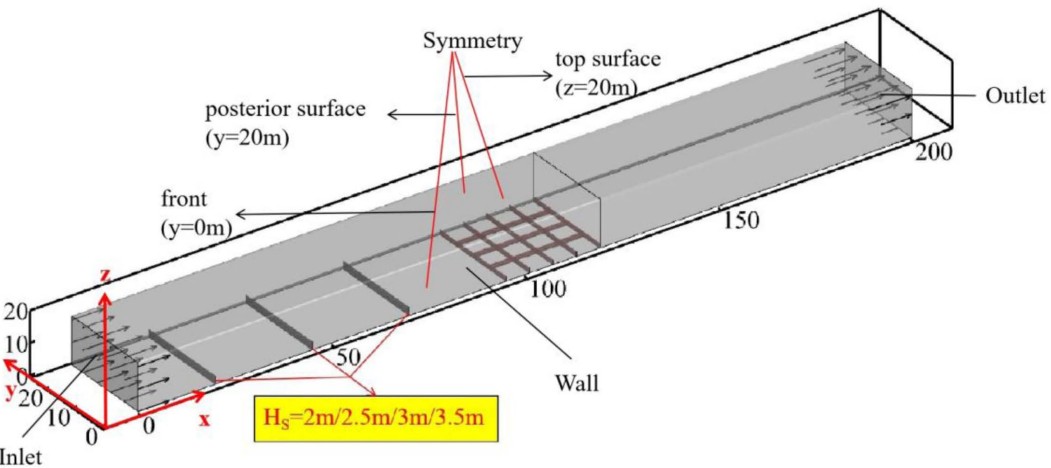

**Fig 2. Schematic diagram of calculation domain of sand barrier fence with different heights.**

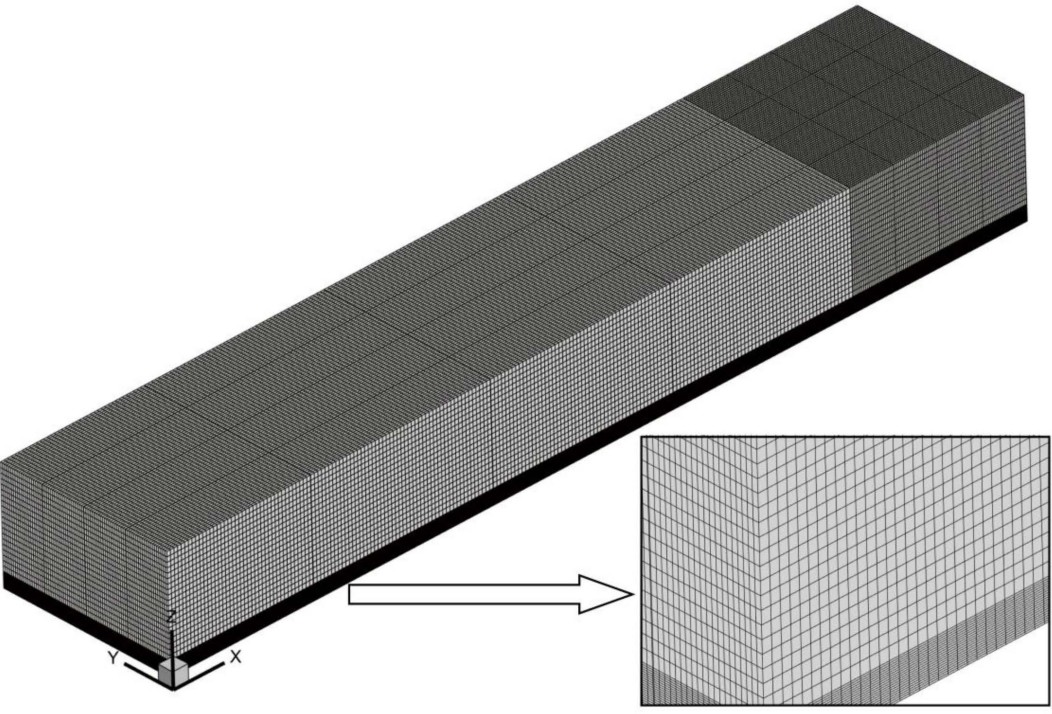

**Fig 3. Grid division diagram.**

used, namely: extracting the horizontal wind speed at the height of 0.1 m on the top surface of the embankment. The division diagram is shown in Fig 4. It can be seen from the figure that the grid division method does not affect the wind speed. The horizontal wind speeds at different positions under the three division methods are almost overlapped, and the wind speed error does not exceed 1%. Therefore, the grid division method does not affect the wind speed. It is proved that the model grid in this paper is irrelevant.

**2.2.3 Governing equations for fluid flow.** The movement of the fluid should follow the law of conservation of physics. The gas phase and the sand phase should conform to the basic continuity equation and satisfy the law of conservation of momentum. Combined with the research object of this paper, the mathematical expression of the two-phase control equation is as follows:

The continuity equation of the gas phase:

$$\frac{\partial}{\partial t}(\alpha_a \rho_a) + \nabla \cdot (\alpha_a \rho_a \vec{v}_a) = 0 \tag{1}$$

The continuity equation of sand phase:

$$\frac{\partial}{\partial t}(\alpha_s \rho_s) + \nabla \cdot (\alpha_s \rho_s \vec{v}_s) = 0 \tag{2}$$

The volume fraction of two phases:

$$\alpha_a + \alpha_s = 1 \tag{3}$$

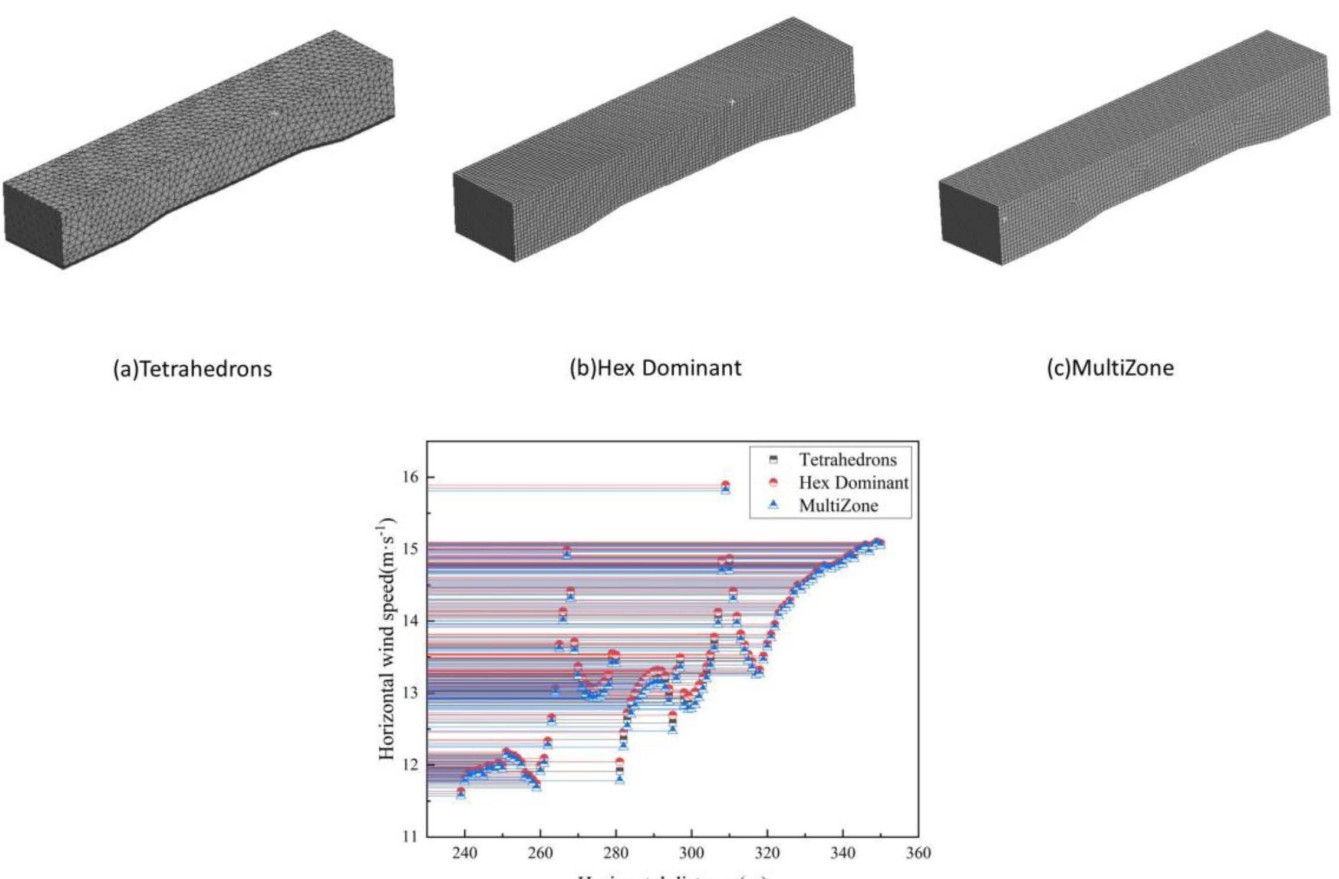

(a)Tetrahedrons   (b)Hex Dominant   (c)MultiZone

(d) Near-surface horizontal wind speed of different grid division methods

**Fig 4. Grid independence verification.**

In the formula: $\rho$—density;

$t$—time;

$v_a$、$v_s$—velocity vectors of gas phase and sand phase;

$\alpha_a$、$\alpha_s$—volume fraction of gas phase and sand phase;

The momentum equation of the gas phase:

$$\frac{\partial}{\partial t}(\alpha_a\rho_a\vec{v}_a) + \nabla\cdot(\alpha_a\rho_a\vec{v}_a\vec{v}_a) = -\alpha_a\nabla p + \nabla\bar{\bar{\tau}}_a + \alpha_a\rho_a\vec{g} + \vec{R}_{pa} + \vec{F}_{lift,a}$$

(4)

The momentum equation of sand phase:

$$\frac{\partial}{\partial t}(\alpha_s\rho_s\vec{v}_s) + \nabla\cdot(\alpha_s\rho_s\vec{v}_s\vec{v}_s) = -\alpha_s\nabla p + \nabla\bar{\bar{\tau}}_s + \alpha_s\rho_s\vec{g} + \vec{R}_{ps} + \vec{F}_{lift,s}$$

(5)

In the formula:: $\vec{g}$ —acceleration of gravity;

$\bar{\bar{\tau}}_a$、$\bar{\bar{\tau}}_s$ —The stress-strain tensor of two phases;

$\vec{F}_{lift,a}$ —lift;

$\vec{R}_{pq}$ —The interaction force between two phases。

$$\overline{\overline{\tau}}_a = \alpha_a \mu_a (\nabla \vec{v}_a + \nabla \vec{v}_a^T) + \alpha_a (\lambda_a - \frac{2}{3}\mu_a)\nabla \cdot \vec{v}_a \overline{I} \tag{6}$$

$$\overline{\overline{\tau}}_s = \alpha_s \mu_s (\nabla \vec{v}_s + \nabla \vec{v}_s^T) + \alpha_s (\lambda_s - \frac{2}{3}\mu_s)\nabla \cdot \vec{v}_s \overline{I} \tag{7}$$

$$\vec{F}_{lift} = -0.5\alpha_s \rho_q (\vec{v}_q - \vec{v}_p) \times (\nabla \times \vec{v}_q) \tag{8}$$

In the formula: $\mu_a$、$\mu_s$ — shear viscosity coefficient;

$\lambda_a$、$\lambda_s$ — bulk viscosity。

**2.2.4 Turbulence modeling.** Since the desert wind-sand flow is generally in a saturated state, the sand volume fraction is large, and the two-fluid model is suitable for simulation. For the Gobi wind-sand flow, it is generally in an unsaturated state, and the sand volume fraction is small, and the Euler-Lagrange method is suitable. At present, most researchers generally use the Euler two-fluid model for desert wind-sand flow [20–26]. Sand particles are mainly affected by gravity, and the wind-sand flow can be regarded as incompressible fluid. The heat exchange between wind-sand flows can be neglected, so the energy equation is not considered.

The Eulerian two-fluid model treats the fluid-solid two-phase material in the computational domain as a continuous medium that penetrates each other. The sum of the two-phase volume fractions is 1, and the gas phase and the sand phase are controlled by their respective mass and momentum conservation equations. The gas-solid two-phase flow satisfies the turbulent transport equation of the standard k-ε model. The specific transport equations of turbulent kinetic energy (k) and turbulent dissipation (ε) are:

$$\frac{\partial}{\partial t}(\rho k) + \frac{\partial}{\partial x_i}(\rho k u_i) = \frac{\partial}{\partial x_j}\left[(\mu + \frac{\mu_t}{\sigma_k})\frac{\partial k}{\partial x_j}\right] + G_k + G_b - \rho\varepsilon - Y_M + S_k \tag{9}$$

$$\frac{\partial}{\partial t}(\rho\varepsilon) + \frac{\partial}{\partial x_i}(\rho\varepsilon u_i) = \frac{\partial}{\partial x_j}\left[(\mu + \frac{\mu_i}{\sigma_\varepsilon})\frac{\partial\varepsilon}{\partial x_j}\right] + C_{1\varepsilon}\frac{\varepsilon}{k}(G_k + C_\mu G_b) - C_{2\varepsilon}\rho\frac{\varepsilon^2}{k} + S_\varepsilon \tag{10}$$

In the formula: $G_k$ -the turbulent kinetic energy generated by the average velocity gradient;

$\mu_i$ – Turbulent (eddy) viscosity When it is incompressible fluid flow and does not consider the user-defined source term, $G_b = 0, Y_M = 0, S_k = 0, S_\varepsilon = 0$.

Turbulent (eddy) viscosity is calculated by the following formula:

$$\mu_t = \rho C_\mu \frac{k^2}{\varepsilon} \tag{11}$$

Here, $\sigma_k$ and $\sigma_\varepsilon$ are the Prandtl constants of turbulent kinetic energy (k) and turbulent dissipation (ε), respectively. $C_{1}$, $C_{2}$ and $C_\mu$ are empirical constants [27]; in this paper, the specific values of the above constants are as follows:

$$C_{1\varepsilon} = 1.44, C_{2\varepsilon} = 1.92, C_\mu = 0.09, \sigma_k = 1.0, \sigma_\varepsilon = 1.3 \tag{12}$$

**2.2.5 Boundary conditions.** In view of the fluidity of sand, the medium type is fluid, the cross section x = 0m is the velocity inlet (VELOCITY-INLET), the cross section x = 200m is the free outlet (OUTLET-FLOW), the bottom surface adopts the wall condition (WALL), and the rest surface adopts the symmetrical boundary condition (SYMMETRY) [28]. The velocity at the entrance of the model in this paper adopts the logarithmic flow wind speed, and the wind speed at the height of 10 m is 15 m/ s. The typical wind speed profile is:

$$\nu\,(z) = \frac{\nu}{k}\,\ln\frac{z}{z_0}$$

(13)

In the formula: $v(z)$-velocity at height $y$;$v$ -friction wind speed;$k$ -Carmen constant, generally 0.4;$z$-the height from the surface;$z_0$-The surface roughness, wind speed is 0 of a certain geometric height, generally take the bed surface sand average particle size of 1/ 30.

The distribution range of sand particle size in the desert section of Wuma Expressway is 0.075–0.25 mm. The sand particle size in the wind-sand flow is set to be 0.1 mm, the sand particle density is 2650 kg/m³, the sand particle kinematic viscosity coefficient is 0.047 kg/(m·s), and the sand particle phase volume fraction is 0.01.

**2.2.6 Reliability verification.** A computational domain with the same size (16m × 1.2m × 1.2m) as the multi-functional environmental wind tunnel experiment of the State Key Laboratory of Desertification and Wind Sand Disaster Prevention and Control in Gansu Province was established. The wind speed at different heights was extracted and compared with the measured wind tunnel data, as shown in Fig 5. It can be seen from the figure that the numerical simulation results of wind speed are basically consistent with the wind tunnel experimental data, which proves that the numerical simulation model and related parameter settings are reasonable.

## 3 Protective benefits of sand barrier at different heights

### 3.1 Flow field structure characteristics of sand barrier area

The cross section of the flow field in the middle (Y = 12.075m) is selected to compare the flow field structure changes of the sand-carrying wind through the sand-blocking fence at different heights. The results are shown in Fig 6. It can be found that the wind-sand flow forms a wide range of low-speed air flow after encountering each sand-blocking fence. However, due to the different heights of the sand-blocking fence, the range of the low-speed air flow is obviously different.

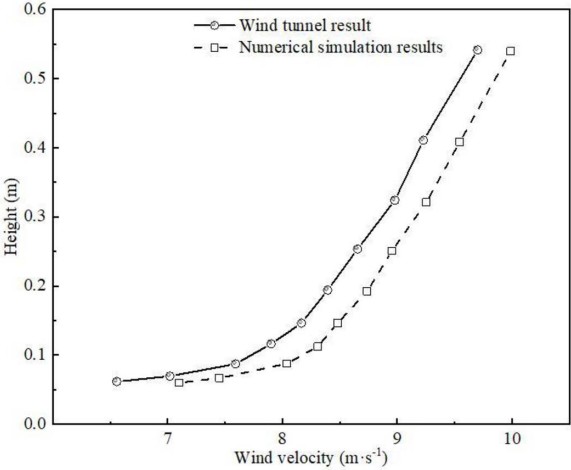

**Fig 5. Comparison of wind speed at different heights in numerical simulation and wind tunnel test.**

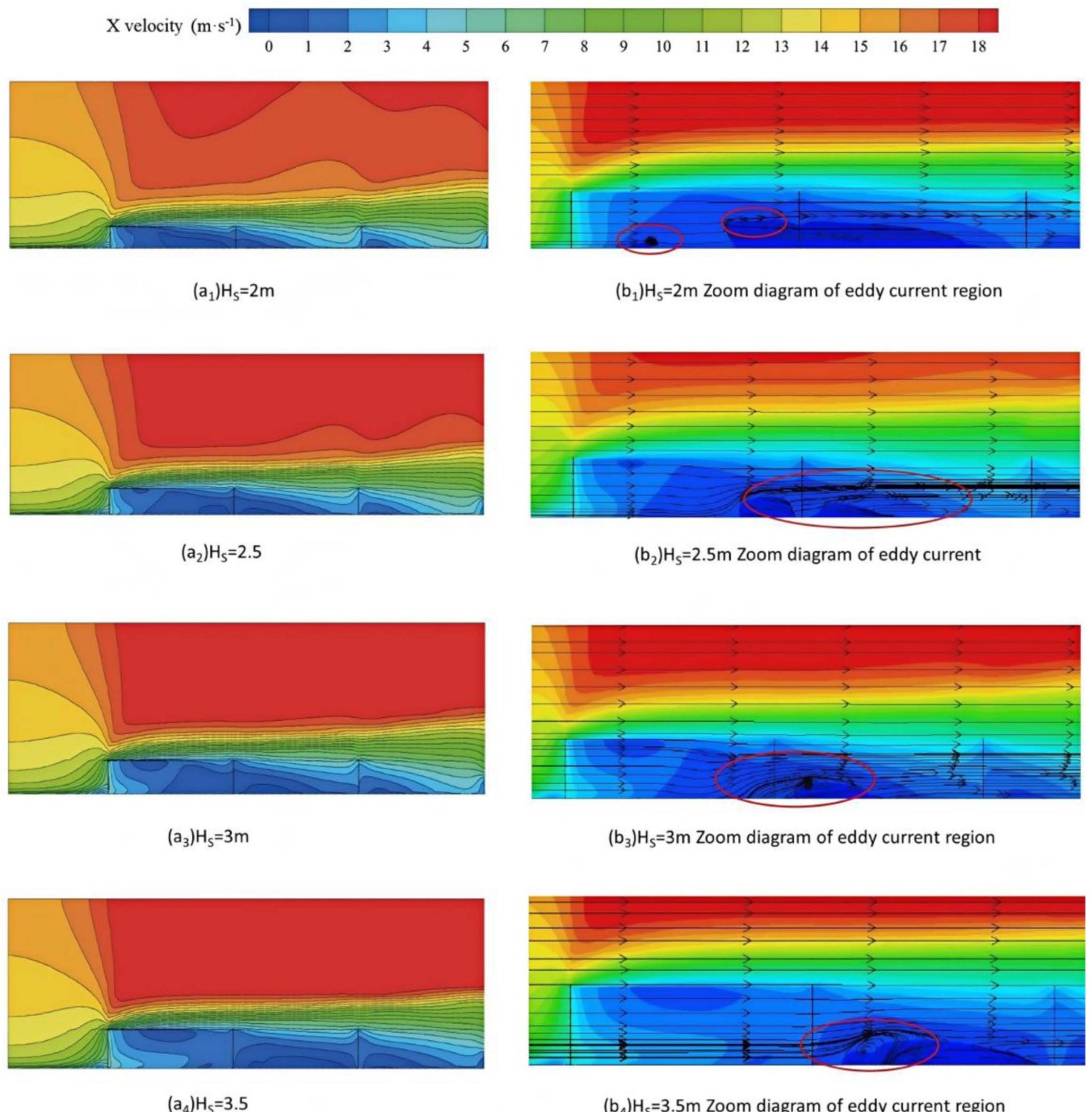

**Fig 6. Flow field structure and local enlargement of eddy current zone under different sand barrier heights.**

It can be seen from Fig 6 (a) that between the first and second sand-blocking fences, the height of the sand-blocking fence increases from 2m to 3m, the area of the low-speed zone of the airflow gradually increases, and the range of the low-speed zone gradually moves backward. However, when the height of the sand-blocking fence increases to 3.5m, the

range of the low-speed zone decreases. The main reason is that the sand-blocking fence is too high, so that the upper airflow velocity is large, the airflow sinking position moves backward, and the airflow passing through the pores is more. The flow field can be fully developed after the first sand-blocking fence. Between the second, third and third vertical large grid sand barriers, with the increase of the height of the sand barrier, the range of the low-speed zone of the airflow gradually increases, and the range of the low-speed zone gradually expands backward.

Fig 6(b) shows the low-speed vortex zones between barriers at different barrier heights (as indicated by the red circle in the figure). From the figure, it can be seen that when the barrier height is 2 m, there is a small-scale vortex zone between the first and second barriers, but no significant vortex zone exists between the second and third barriers. When the barrier height is lower, the airflow experiences less kinetic energy loss after passing the first barrier, leading to faster wind speed recovery. After the airflow sinks behind the first barrier, some of its energy remains sufficient to maintain flow, thus forming a small-scale vortex zone (local kinetic energy dissipation) between the first and second barriers. However, when the airflow reaches the second and third barriers, the remaining kinetic energy is insufficient to cause significant separation, so there is no obvious vortex zone between the second and third barriers (energy has been sufficiently attenuated, and the airflow tends to become advection);When the height of the sand-blocking fence is 2.5 m, significant vortex zones appear between the first and second sand-blocking fences, as well as between the second and third sand-blocking fences. After increasing the fence height, the kinetic energy loss when the airflow passes over the first fence becomes more pronounced, and the downward airflow velocity decreases more noticeably, leading to stronger backflow (larger low-pressure areas) behind the first fence. The obstruction of the second fence further intensifies airflow separation, creating even larger vortex zones. At this point, the spacing and height of the fences match, allowing the airflow to complete an "acceleration-deceleration-separation" cycle between the two fences, maximizing the range of the vortex zone. When the height of the sand-blocking fence is 3m and 3.5 m, the range of the vortex zone gradually decreases. This is because excessively high fences cause intense turbulence when the airflow passes over them, leading to rapid energy dissipation near the top of the fence, significantly reducing the downward airflow velocity and weakening the subsequent airflow disturbance between the fences.

### 3.2 Windproof effect of sand barrier fences with different heights

**3.2.1 Horizontal wind speed.** The wind-sand flow is generally close to the surface movement. The higher the sand barrier is, the more the wind-sand flow can be blocked, and the weakening effect on the near-surface wind speed is enhanced, thus improving the sand blocking effect. Therefore, the study of near-surface wind speed changes can clarify the blocking effect of protective measures on wind-sand flow [29,30]. The horizontal wind speed at the height of 0.1m and 0.3m from the surface of the three sand-blocking fences at different heights is shown in Fig 7.

It can be seen from Fig 7 that at a distance of 0.1 m from the surface, the airflow velocity drops sharply after encountering the sand barrier. When the height of the sand barrier is 2m, 2.5m, 3m, and 3.5m, the airflow velocity decreases between the first and second sand barriers. The initial wind speed is 87.81%, 94.90%, 92.33%, and 89.21%; between the second and third sand-blocking fences, the airflow velocity experienced a slight decrease and then gradually increased. The increase of airflow velocity was 36.82%, 24.62%, 19.83% and 11.74% of the initial wind speed, respectively. At 0.3 m from the surface, the airflow velocity decreases by 92.08%, 96.64%, 94.07% and 90.71% of the initial wind speed, respectively. Between the second and third sand-blocking fences, the airflow velocity experienced a slight decrease and then gradually increased. The increase of airflow velocity was 36.61%, 22.74%, 18.52% and 10.78% of the initial wind speed, respectively.

**3.2.2 Windproof efficiency.** Windbreak efficiency is an important index to measure the blocking effect of protective measures on wind-sand flow. It reflects the ability of protective measures to inhibit wind-sand flow and prevent sand migration. The larger the value is, the stronger the blocking effect is. The windbreak efficiency can be calculated by the following formula [31].

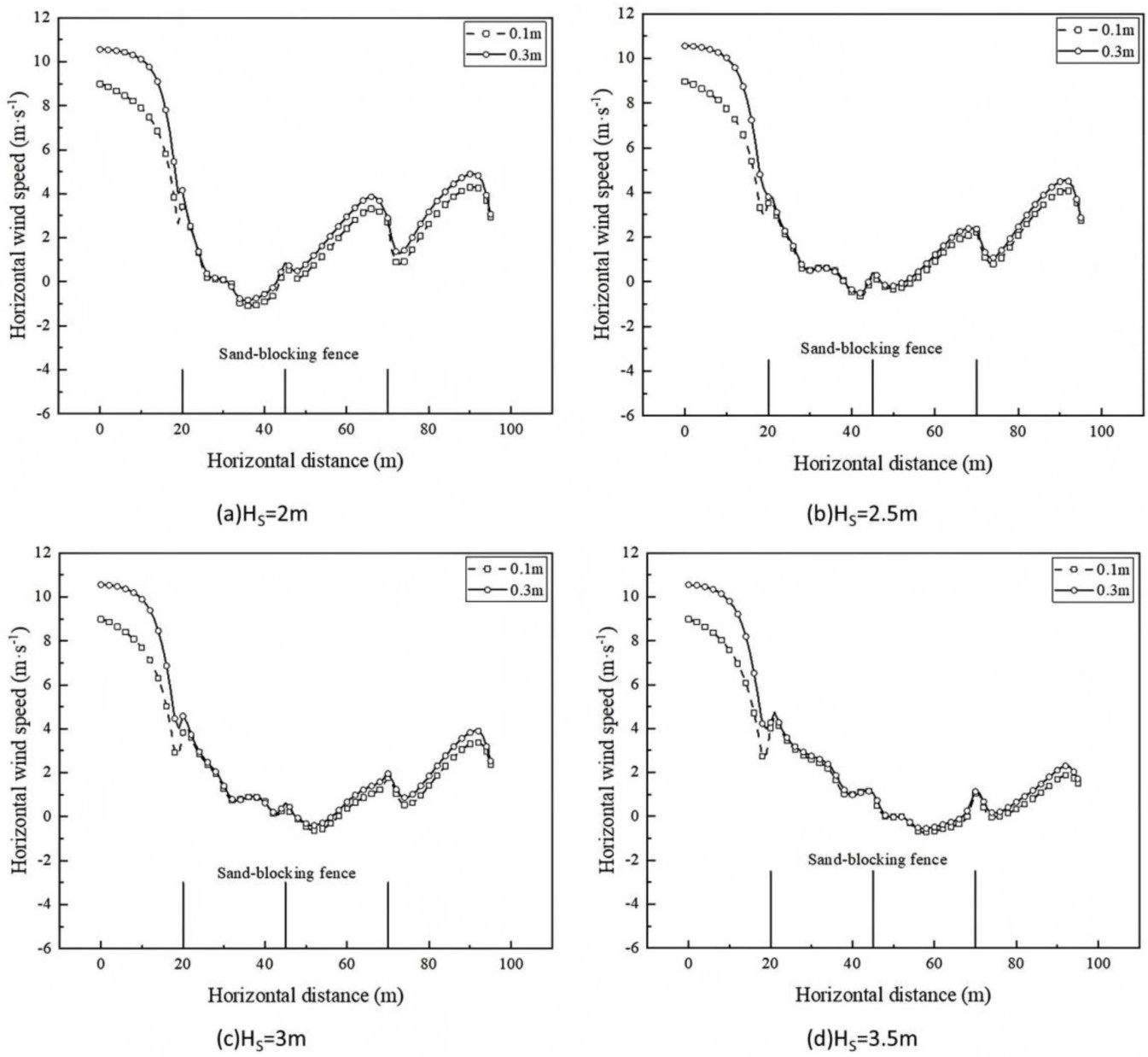

**Fig 7. Horizontal wind speed changes of sand-blocking measures at different heights.**

$$\Phi_{xz} = \left(1 - \frac{V_{xz}}{V'_{xz}}\right) \times 100\%$$

(14)

Among them, $\Phi_{xz}$ is the windproof efficiency (the maximum windproof efficiency is 100%); x is the length from the first sand barrier; z is the height from the ground, the unit is m; $V'_{xz}$ is the wind speed at the point (x, z) in m/s when there are protective measures; $V'_{xz}$ is the wind speed at point (x, z) without protective measures, and the unit is m/s.

The wind speed at the height of 0.1m and 0.3m from the surface in the range of three sand barrier fences is extracted, and the wind protection efficiency is calculated. The results are shown in . At the height of 0.1m from the surface, when the height of the sand barrier fence is 2m, 2.5m, 3m and 3.5m, the average wind-proof efficiency between the first and second sand barrier fences is 89.83%, 86.94%, 81.55% and 71.82%, respectively. The average wind-proof efficiency between the second and third sand barrier fences is 79.74%, 90.22%, 93.16% and 95.26%, respectively. At a height of 0.3 m from the surface, under the conditions of different heights of sand-blocking fences, the average wind-proof efficiency between the first and second sand-blocking fences is 92.35%, 89.10%, 84.08% and 75.31%, respectively. The average

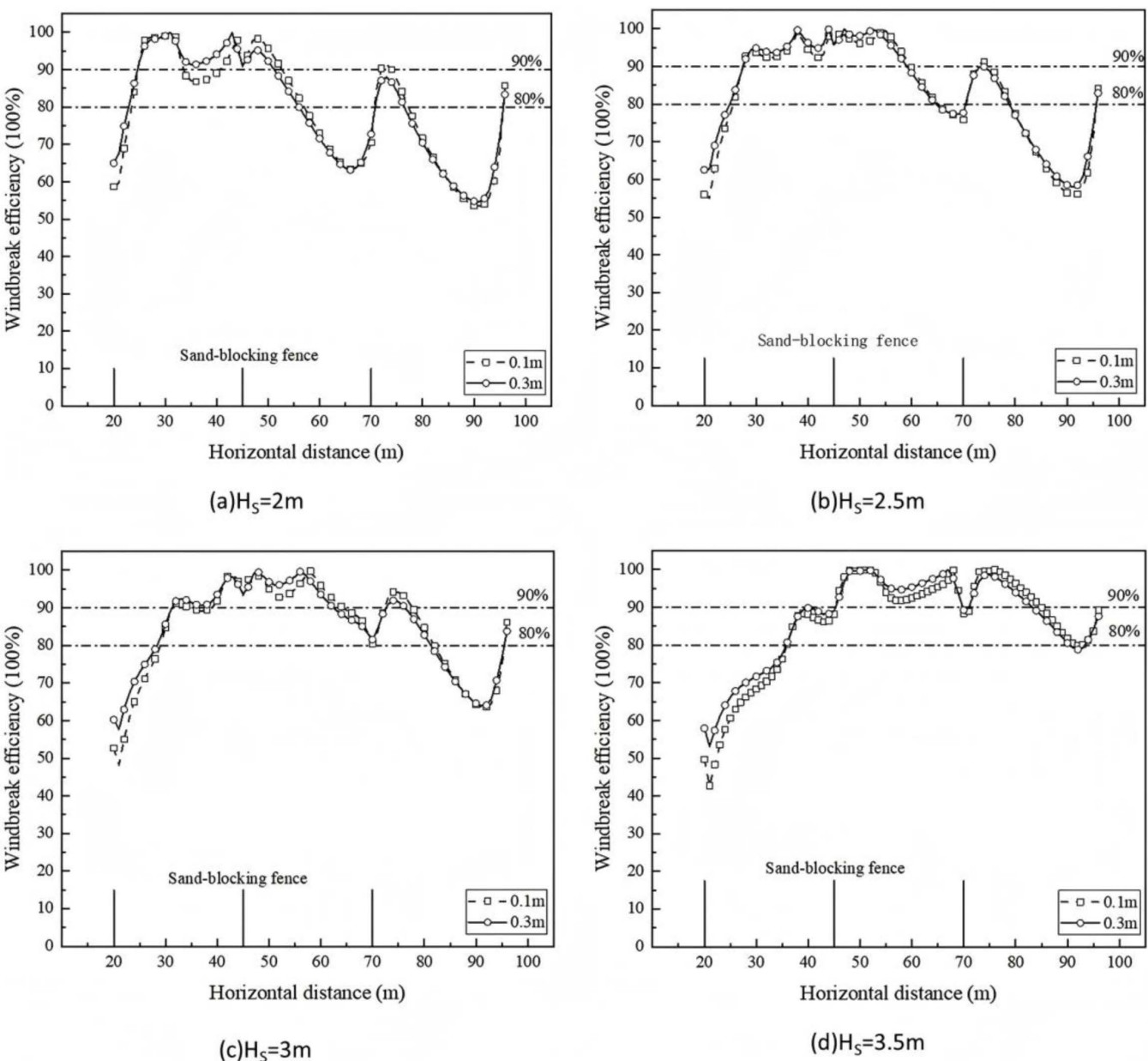

**Fig 8. Windproof efficiency of sand barrier at different heights.**

wind-proof efficiency between the second and third sand-blocking fences is 78.26%, 90.16%, 93.13% and 96.44%, respectively.

### 3.3 Sand-blocking effect of different heights of sand-blocking fence

The three-dimensional distribution cloud images of sand accumulation, sand coverage, and sand volume fraction at different heights of the sand barriers were extracted for a sand thickness of 0.55 to 0.6 meters, as shown in Fig 8. From the Fig 9, it can be seen that when the height of the sand barrier is 2 meters, the sand-blocking range basically covers the entire interval between the first and second barriers, with only a very small area remaining uncovered after the first barrier. When the height of the sand barrier is 2.5 meters, the sand accumulation range significantly expands, covering the entire interval with an average thickness of 0.295 meters, significantly enhancing the sand prevention effect. Between the second and third barriers, the sand accumulation thicknesses at different heights are 0.096 m, 0.152 m, 0.155 m, and 0.154 m, When the height exceeds 3m, the sand accumulation range between the two and three sand barriers is reduced, and the sand prevention effect is weakened.

## 4 Discussions

### 4.1 Comparison of wind and sand prevention effects of sand barrier fences with different heights

Table 1 shows the variation range of near-surface horizontal wind speed and the proportion of initial wind speed at different heights of sand-blocking fences. It can be seen from the table that the horizontal wind speed decreases by about

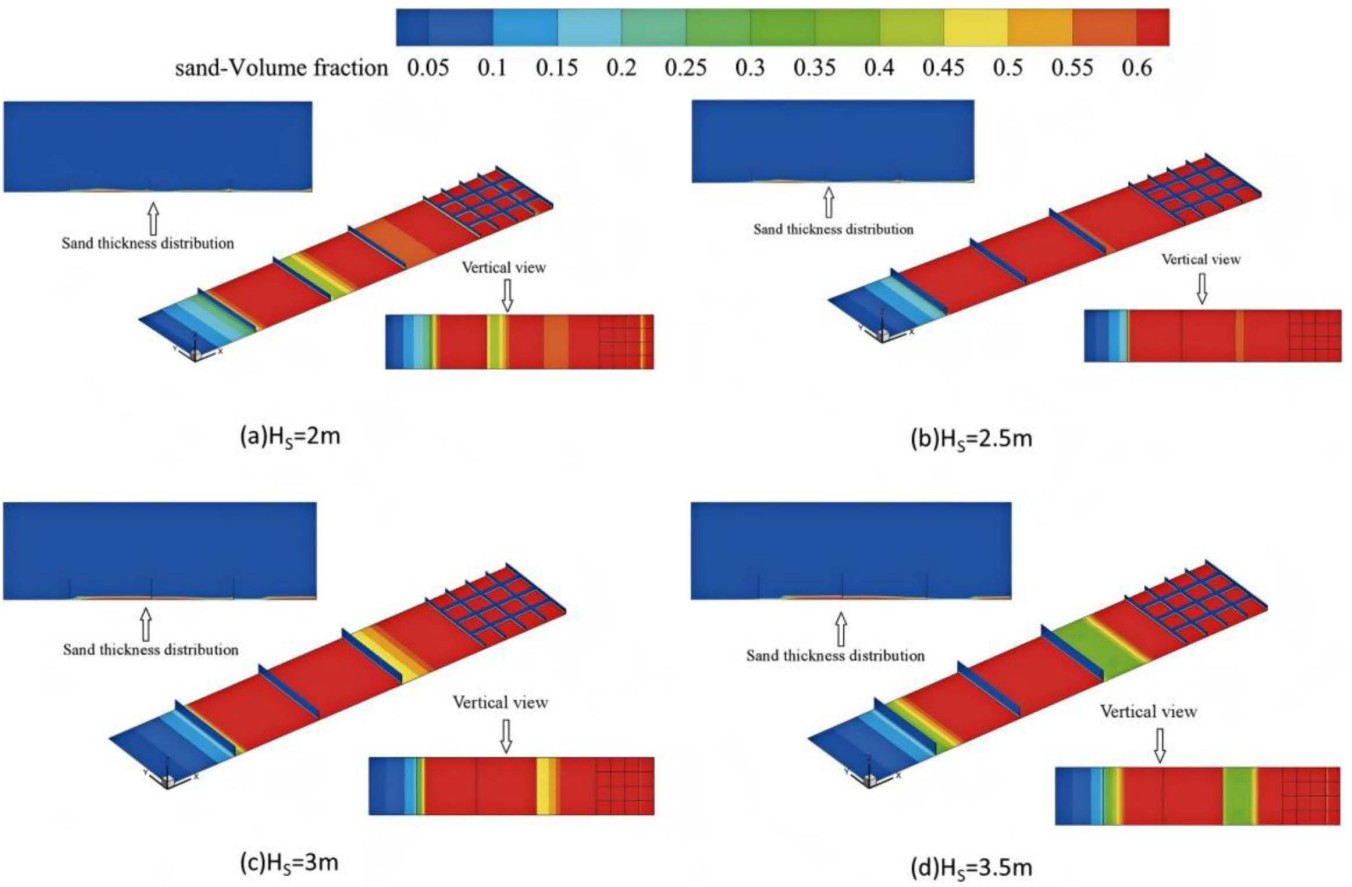

**Fig 9. The sand cloud around the sand barrier at different heights.**

Table 1. The change range of wind speed and the proportion of initial wind speed at each position of sand barrier before and after improvement.

| Height of sand barrier (m) | The velocity of first and second sand-blocking fences (%) | | The velocity of Second and third sand-blocking fences (%) | |
|---|---|---|---|---|
| | 0.1m | 0.3m | 0.1m | 0.3m |
| 2 | 87.81 | 92.08 | 36.82 | 36.61 |
| 2.5 | 94.90 | 96.64 | 24.62 | 22.74 |
| 3 | 92.33 | 94.07 | 19.83 | 18.52 |
| 3.5 | 89.21 | 90.71 | 11.74 | 10.78 |

87% −97% of the initial wind speed between the first and second sand-blocking fences, and a large number of sand grains are deposited here. When the height of sand-blocking fence is 2.5 m, the velocity decreases the most, which is 4.56% higher than that of 2 m. The velocity decreases at 3 m and 3.5 m are higher than that of 2.5 m. The main reason is that the sand-blocking fence is too high, which makes the upper airflow velocity larger, the airflow sinking position moves backward, and the airflow passing through the pores is more. The flow field can be fully developed after the first sand barrier; between the second and third sand-blocking fences, with the increase of the height of the sand-blocking fence, the increase of airflow velocity gradually decreases. When the height of the sand-blocking fence is 2.5m, the increase of wind speed is 13.87% lower than that of the previous level, and the decrease is the largest. The recovery of airflow velocity is the slowest and cannot be fully developed.

Table 2 is the calculated average wind-proof efficiency at each position of different sand-blocking fence heights. From the table, it can be seen that between the first and second sand-blocking fences, when the height of the sand-blocking fence is 2m, 2.5m, and 3m, the wind-proof efficiency is slightly reduced, but it is around 80%～90%. When the height increases to 3.5m, the average wind-proof efficiency is greatly reduced; between the second and third sand-blocking fences, when the height is 2.5m, the wind-proof efficiency is 11.9% higher than that of 2m, and the wind-proof effect is greatly improved. After that, when the height is increased to 3.5m, the wind-proof efficiency is above 90%, and the change range tends to be stable. Obviously, the increase of height has a more significant effect on the wind protection efficiency between the second and third sand barrier fences.

It can be seen from the table that when the height of the sand-blocking fence is 2m, the wind-proof efficiency between the second and third sand-blocking fences is the largest than that between the first and second sand-blocking fences, which is reduced by 14.09%. When the height of the sand-blocking fence is 2.5m, the wind-proof efficiency between the second and third sand-blocking fences increases, and the wind-proof efficiency between the first and second sand-blocking fences does not decrease significantly.

When the height of the sand-blocking fence is 3m, the increase in wind-proof efficiency between the second and third sand-blocking fences is not much different from that of the sand-blocking fence 2.5m, but the wind-proof efficiency

Table 2. The average windproof efficiency of each position of sand barrier before and after improvement.

| Height of sand barrier (m) | The average windbreak efficiency of the first to the second sand barrier (%) | | The average windbreak efficiency of the second to the third sand barrier (%) | |
|---|---|---|---|---|
| | 0.1m | 0.3m | 0.1m | 0.3m |
| 2 | 89.83 | 92.35 | 79.74 | 78.26 |
| 2.5 | 86.94 | 89.10 | 90.22 | 90.16 |
| 3 | 81.55 | 84.08 | 93.16 | 93.13 |
| 3.5 | 71.82 | 75.31 | 95.26 | 96.44 |

between the first and second sand-blocking fences decreases. When the height of the sand-blocking fence is 3.5m, the wind-proof efficiency between the first and second sand-blocking fences decreases the most, and the wind-proof effect is not fully exerted. Therefore, in summary, compared with other heights, when the height of the sand-blocking fence is increased to 2.5m, it has a better wind-proof and sand-blocking effect.

The thickness of sand accumulation in the range of 0.55~0.6 is extracted from the sand volume fraction between three different heights of sand-blocking fences. The results are shown in Table 3. It can be seen from the table that the thickness of sand accumulation increases with the height of sand-blocking fence between the first and second sand-blocking fences and between the second and third sand-blocking fences. When the height of sand-blocking fence is 2.5m, the thickness of sand accumulation is 50.51% and 58.33% higher than that of 2m high sand-blocking fence. It has a better blocking effect on sand particles in the airflow, and the effect of sand-blocking is the most significant. When the height increases to 3.5m, the thickness of sand accumulation does not increase significantly.

It can be seen that with the increase of the height of the sand-blocking fence, the thickness of the sand between the first and second sand-blocking fences and between the second and third sand-blocking fences increased compared with the height of the sand-blocking fence at the site of 2m. When the height of the sand-blocking fence increases to more than 2.5m, the thickness of the sand reaches a saturated state and no longer increases. It can be seen that the height of the sand-blocking fence should not be too low. Too low will lead to the inability to effectively block the deposition of sand particles and then move forward, which will cause sand burial and other phenomena. The high windproof efficiency of the sand barrier has not been greatly improved, and it will bring the risk of being blown down by the strong wind while increasing the cost. In summary, when the height of the sand barrier is increased to 2.5 m, it is the most reasonable height and has a better wind and sand resistance effect.

## 4.2 Protective effect verification

In order to verify the protective effect of the recommended height value on the high-risk section of Wuma Expressway, the height of the sand barrier in the original section protection measures is adjusted to 2.5m, and then other original protection measures and roadbed are added (roadbed height H = 3m, slope rate is 1: 4), and the corresponding model is established as shown in Fig 10.

The sand volume fraction cloud map of embankment under 2m and 2.5m protection system is extracted. The results are shown in Fig 11. The darker the color is, the larger the sand volume fraction is, and the easier the sand is to form deposition. From the diagram, it can be seen that when the height of the sand barrier is 2m, the sand volume fraction on the top of the embankment is larger in the range of 0.5~0.6, and when the height of the sand barrier is 2.5m, the sand range on the top of the embankment is significantly reduced, which is 73.44% lower than that of the height of the sand

Table 3. The thickness of sand accumulation in the sand barrier fence is 0.55~0.6 thickness of sand accumulation in the color belt.

| Height of sand barrier (m) | The Sand thickness of first and second sand-blocking fences (m) | The Sand thickness of second and third sand-blocking fences (m) |
|---|---|---|
| 2 | 0.196 | 0.096 |
| 2.5 | 0.295 | 0.152 |
| 3 | 0.295 | 0.155 |
| 3.5 | 0.294 | 0.154 |

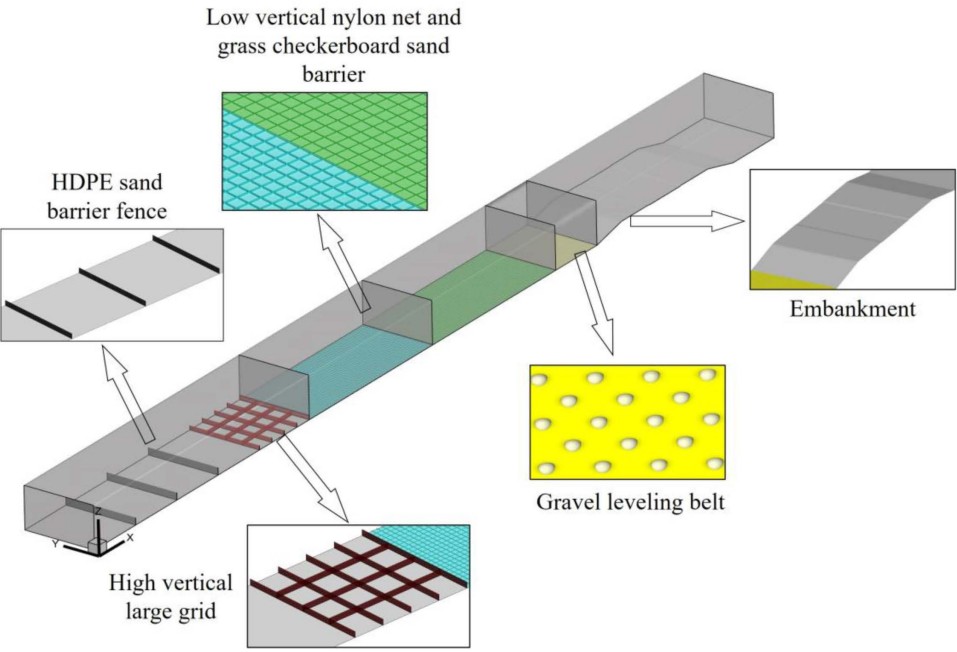

**Fig 10. Model calculation diagram.**

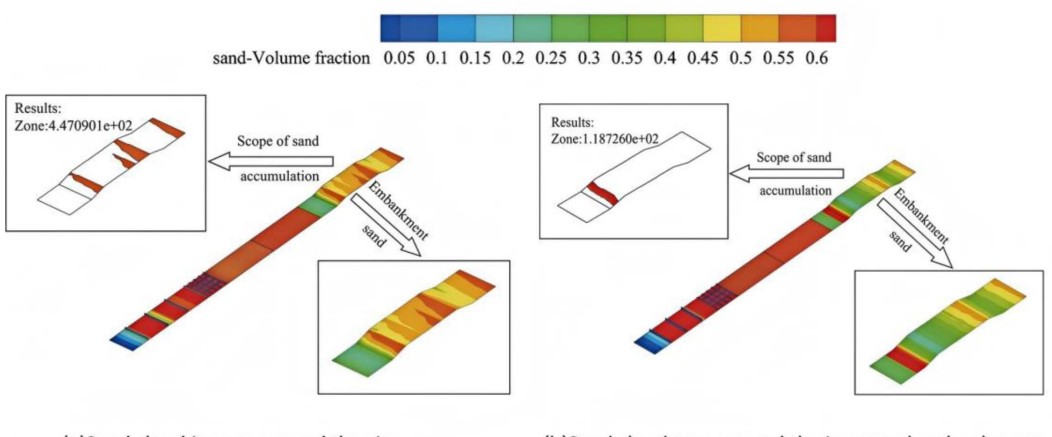

**Fig 11. Sand accumulation nephogram of embankment before and after improvement of sand barrier fence.**

barrier 2m. It can be seen that when the height of the sand barrier is 2.5m, the sand blocking effect is enhanced and the protection effect is greatly improved.

## 5 Conclusion

(1) In the first and second sand-blocking fences, the velocity drop is about 87% −97% of the initial wind speed, and a large amount of sand is deposited here. When the height of the sand-blocking fence is 2.5 m, the velocity drop is the

largest, which is 4.56% higher than that of the height of 2 m. The velocity drop of 3 m and 3.5 m is higher than that of 2.5 m. In the second and third sand-blocking fences, with the increase of the height of the sand-blocking fence, the increase of airflow velocity decreases gradually. When the height of the sand-blocking fence is 2.5 m, the increase of wind speed is 13.87% lower than that of the previous height, and the decrease is the largest. The recovery of airflow velocity is the slowest and cannot be fully developed. A large amount of sand is deposited here.

(2) When the height is 2.5m, the windproof efficiency is 11.9% higher than that of 2m, and the windproof effect is greatly improved. When the height is increased to 3.5m, the windproof efficiency is above 90%, and the change range tends to be stable.

(3) Between the first and second sand-blocking fences and between the second and third sand-blocking fences, the thickness of accumulated sand increases with the height of sand-blocking fence. When the height of sand-blocking fence is 2.5m, the thickness of accumulated sand increases by 50.51% and 58.33% compared with that of 2m high sand-blocking fence, which has a better blocking effect on sand particles in the airflow, and the effect of sand-blocking is the most significant. When the height increases to 3.5 m, the thickness of the sediment does not increase significantly.

(4) When the height of sand barrier is 2.5m, the sand deposition range completely covers the entire sand accumulation area. Compared with the sand accumulation in the embankment section before improvement, the sand range on the embankment top area is significantly reduced. The area with sand volume fraction of 0.55–0.6 is reduced by 73.44% compared with the height of sand barrier of 2m, which has good wind and sand resistance effect.

## Author contributions

**Conceptualization:** Ming Zhang, Qi Li.

**Formal analysis:** Ming Zhang, Qi Li, Jiayin Hou.

**Investigation:** Jiayin Hou.

**Validation:** Shuai Ji.

**Visualization:** Shuai Ji.

**Writing – original draft:** Ming Zhang.

**Writing – review & editing:** Qi Li.

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
