## [Decision Letter · Decision Letter 0]

29 Sep 2024

Dear Dr. Li,

Thank you for submitting your manuscript to PLOS ONE. After careful consideration, we feel that it has merit but does not fully meet PLOS ONE’s publication criteria as it currently stands. Therefore, we invite you to submit a revised version of the manuscript that addresses the points raised during the review process.

The authors need to consider all the suggestions of the reviewers such as including relevant references, providing details/justification of the boundary conditions and grid chosen, explanation for the variation in efficiencies with height in the revised paper.

All the equations and symbols should also be checked

We look forward to receiving your revised manuscript.

Kind regards,

Muhammad Shakaib, PhD

Academic Editor

PLOS ONE

Journal Requirements:

2.Thank you for stating the following financial disclosure: 

 Ningxia Transportation Department Science and Technology Project (20200173);Ningxia key research and development plan project support(2021BEG02017).  

The research described in this paper was financially supported by Youth Science Foundation Project’ Research on

Failure Mechanism and Evaluation Method of Sand Control Measures for Railway Machinery in Sandy Area’ (12302511) ;

Ningxia Transportation Department Science and Technology Project (20200173) ; Central guide local science and

technology development funds (22ZY1QA005).

Ningxia Transportation Department Science and Technology Project (20200173);Ningxia key research and development plan project support(2021BEG02017) 

5. We note that Figures 1, 2, and 3 in your submission contain [map/satellite] images which may be copyrighted. All PLOS content is published under the Creative Commons Attribution License (CC BY 4.0), which means that the manuscript, images, and Supporting Information files will be freely available online, and any third party is permitted to access, download, copy, distribute, and use these materials in any way, even commercially, with proper attribution. For these reasons, we cannot publish previously copyrighted maps or satellite images created using proprietary data, such as Google software (Google Maps, Street View, and Earth). For more information, see our copyright guidelines: http://journals.plos.org/plosone/s/licenses-and-copyright.

a. You may seek permission from the original copyright holder of Figures 1, 2, and 3 to publish the content specifically under the CC BY 4.0 license.  

6. Figures should not be in both the manuscript, and as separate files.

Reviewers' comments:

Reviewer's Responses to Questions

**Comments to the Author**

1. Is the manuscript technically sound, and do the data support the conclusions?

Reviewer #1: Partly

Reviewer #2: Partly

2. Has the statistical analysis been performed appropriately and rigorously?

Reviewer #1: N/A

Reviewer #2: N/A

3. Have the authors made all data underlying the findings in their manuscript fully available?

Reviewer #1: Yes

Reviewer #2: Yes

4. Is the manuscript presented in an intelligible fashion and written in standard English?

Reviewer #1: Yes

Reviewer #2: No

Reviewer #1: Referee-Report: PONE-D-24-35646

Title: Evaluation of protection benefit of HDPE sand barrier fence with different heights on desert Highway

The paper focuses on a CFD numerical simulation model to investigate the wind and sand blocking effects of fences with varying heights and constant porosity. The authors utilize ANSYS FLUENT software to implement a two-phase Euler model, employing the SIMPLEC (Semi-Implicit Method for Pressure-Linked Equations – Consistent) algorithm. The simulation results demonstrate that fence barriers with different heights have a significant impact on reducing wind-driven sand transport. These findings are particularly relevant for protecting a section of the Wuma Expressway, located in the Tengger Desert.

The paper presents numerical results on an interesting topic related to wind-driven sand transport. The main objective is to demonstrate how varying fence heights can improve protection or reduce sand transport. While the simulation box used in their model is well described, there is a lack of clarity regarding the choice of boundary conditions for the fluid flow, particularly the decision to use a symmetric boundary at the top of the simulation box. The simulation methods themselves also require more detailed explanation to make the approach fully understandable to readers. Although the authors present results confirming that fences can be effective in reducing sand transport, they do not engage in a deeper scientific exploration of the results. The paper mainly describes the findings from the simulations without critically analyzing or discussing them. For example, there is no discussion on optimization strategies for determining the most effective fence height for protection. Is there a specific rationale for choosing certain fence heights, or does height selection not significantly impact performance? This aspect needs further exploration to provide practical insights for those interested in applying these findings to real-world scenarios.

The manuscript presents a variety of simulations and results that support the authors' conclusions; however, it lacks the necessary discussion to emphasize and critically analyze the findings. Due to these concerns, the manuscript is not suitable for publication in its current form. There are several points that require clarification, which I outline below.

1) There are some abbreviations that must be translate (UAV) and (HDPE).

2) The simulation grid description is very confused. They should make clear for reader. The images are good, but the description is not.

3) Did the authors test the mesh quality, or was the mesh chosen primarily based on simulation time considerations? Additionally, there appears to be some confusion between the terms "grid" and "cell." Clarification and consistency in the usage of these terms are needed.

4) There is insufficient description of the conditions under which the fluid and particles interact. While this may be implicit in the CFD method, it is important to clearly define these conditions. How do the authors define sand accumulation, particularly if the model used is based on a two-phase approach?

5) The Authors comment that “Desert wind-sand flow is generally in a saturated state”, but they do not explain what it means or use references about the subject.

6) Is there a specific reason for using a symmetric boundary condition at the top? To be honest, I did not fully understand the sentence: “the rest surface adopts the symmetrical boundary conditions” (SYMMETRY). Could the authors clarify this point?

7) I am confused about the coordinate’s description in eq. (5) (input velocity profile).

8) Could the authors provide a 3D view of the velocity field? I am particularly interested in observing the velocity field near the fence borders.

9) In the section on horizontal wind speed, it is unclear whether the velocity is composed of two components along the z and x directions or if it is only a function of x. Additionally, the results presented in this section show comparative values, but it is not clear whether these values are related to the velocity field without fences or if they are being compared to the input profile.

10) The velocity field profiles (contour lines) shown in Figure 7 appear almost identical, with only slight differences. How, then, could the calculated efficiency values differ so significantly, given that they are derived from the velocity profile? Is there an explanation for this apparent discrepancy?

11) Table 2 raises the question of whether there are any correlations between the efficiency of the first/second barrier and the second/third barrier. For which fence height is the decrease in efficiency more pronounced? Additionally, does this depend on the distance from the bottom surface?

12) The authors should provide a more thorough analysis of the data presented in the tables. For instance, the data in Table 3 indicates that accumulation reaches saturation as the fence height increases. However, no discussion of these results is included in the paper.

13) Several sentences need to be revised for clarity, as they may confuse readers.

14) The authors should review all figure captions to enhance their descriptions, as captions are essential for understanding the figures.

15) We know that fence efficiency is measure in terms of transport. I think the Authors should include a conection with sand transport in terms of the shear stress treshold for the sand movment. The shear treshold depen on the grain size. There are some discussions how to introduce this connection on the following papers:

Sauermann, G., Andrade Jr., J.S., Maia, L.P., Costa, U.M.S., Araújo, A.D., Herrmann, H.J., 2003. Wind velocity and sand transport on a barchan dune. Geomorphology 54,

245–255.

Guan, D.X., Zhong, Y., Jin, C.J., Wang, A.Z., Wu, J.B., Shi, T.T., Zhu, T.Y., 2009. Variation in wind speed and surface shear stress from open floor to porous parallel windbreaks: a wind tunnel study. J. Geophys. Res. 114, D15106.

Lima, I. A., Parteli, E. J., Shao, Y., Andrade, J. S., Herrmann, H. J., & Araújo, A. D. (2020). CFD simulation of the wind field over a terrain with sand fences: Critical spacing for the wind shear velocity. Aeolian Research, 43, 100574.

Cornelis, W.M., Gabriels, D., 2005. Optimal windbreak design for wind-erosion control. J.Arid Environ. 61, 315–332.

Reviewer #2: This manuscript presents a numerical simulation study of the effect of wind barriers on sand transport using Computational Fluid Dynamics. IT focuses on the effect of fences designed to reduce sandblasting in an area known as Wuma Expressway, which passes through the hinterland of the Tengger Desert from east to southwest. This type of study is very important for desertification, climate and agricultural sciences, and we need more models, experiments and numerical simulations to better elucidate the physics of soil erosion and prevention.

However, in the present form, I cannot recommend publication of this manuscript in Plos ONE for the reasons mentioned below. If the authors satisfactorily revise their manuscript by taking all points below into account though, then I would be glad to participate as reviewer again in a future review round of this paper.

--------------

COMMENT 1)

--------------

The introduction and conclusions read like a report for the Chinese government. The text mentions a lot of numbers for sand density, porosity, wind speed, etc., specifically about the model for the study area, without a clear message about what is being learned from the simulation results. Moreover, are there any insights from the simulation that could be maybe useful for combating desertification or soil erosion in other areas of the world? Overall, this paper reads like a governmental report, rather than a scientific paper.

--------------

COMMENT 2)

--------------

One more piece of evidence that this has been a governmental report, which has been translated to English and then submitted to Plos ONE: The bibliography contains ONLY publications from Chinese authors. However, there is so much groundbreaking literature on CFD modeling of wind-blown sand transport, completely ignored by the authors. For example, Smyth (2016) wrote an outstanding review on CFD applied to this area, see here:

https://doi.org/10.1016/j.aeolia.2016.07.003  (Smyth, 2016)

Moreover, Lima et al. (2017) and Lima et al. (2020) published similar numerical simulations to the ones presented here, thereby deriving models for fence efficiency as a function of porosity, height and spacing, see the links to the two very important papers here:

https://www.nature.com/articles/srep45148  (Lima et al., 2017)

https://doi.org/10.1016/j.aeolia.2020.100574   (Lima et al., 2020)

All these papers should be definitely incorporated into the bibliography, and the authors should diversify a little more the literature in general to include more authors from other countries but China.

--------------

COMMENT 3)

--------------

The paper overall is not well written. A thorough revision of the language is definitely required. Moreover, the model description has various flaws. I skipped some parts of it since I will only give any further consideration to this paper if the authors resubmit a version that successfully addresses the major comments raised here, which are enough to preclude publication in PLOS ONE. However, for example, equation (5) is described incorrectly. The height above the ground is "z" and not "y". Moreover, "y_0" is not the wind speed at surface, it is rather the roughness, which is the height at which the wind velocity is zero. This kind of error shows one thing: The authors did not perform a careful job. They just submitted this manuscript to see what happens. And it astounds me that the journal chosen by the authors is PLOS ONE. This is a strong interdisciplinary journal, to which we should submit papers of high quality only.

Therefore, my final recommendation is rejection. But I am opting for "Major Revision" in the electronic system to signalize to the editorial office that I would agree to review an improved version along the lines above. The authors must give the paper to someone who can help them check the correctness of the equations and plots, because even the simplest equation (Eq. (5)) was presented incorrectly.

**Do you want your identity to be public for this peer review?** For information about this choice, including consent withdrawal, please see our Privacy Policy

Reviewer #1: No

Reviewer #2: No

---

## [Author Response · Author response to Decision Letter 1]

18 Dec 2024

Dear editor:

First of all, thank you very much for taking the time to read and evaluate my manuscript carefully in your busy work. Your valuable comments and suggestions are of great significance to me. Secondly, in your letter, I noticed that you have put forward some opinions and suggestions on the format of my manuscript. I have further modified the format of the paper to make it conform to the requirements of the format of your paper. At the same time, if there are other problems in the paper, I hope you can tell me that I will modify and reply in time, and hope that the manuscript can be published in your journal as soon as possible. Finally, I wish you and the review experts a happy life and a smooth work ! Thank you again !

---

## [Decision Letter · Decision Letter 1]

5 Jan 2025

Dear Dr. Li,

Thank you for submitting your manuscript to PLOS ONE. After careful consideration, we feel that it has merit but does not fully meet PLOS ONE’s publication criteria as it currently stands. Therefore, we invite you to submit a revised version of the manuscript that addresses the points raised during the review process.

The authors are suggested to carefully consider the previous comments of reviewer 2 and incorporate in the revised paper.

We look forward to receiving your revised manuscript.

Kind regards,

Muhammad Shakaib, PhD

Academic Editor

PLOS ONE

Reviewers' comments:

Reviewer's Responses to Questions

**Comments to the Author**

Reviewer #2: (No Response)

2. Is the manuscript technically sound, and do the data support the conclusions?

Reviewer #2: Partly

3. Has the statistical analysis been performed appropriately and rigorously?

Reviewer #2: N/A

4. Have the authors made all data underlying the findings in their manuscript fully available?

Reviewer #2: Yes

5. Is the manuscript presented in an intelligible fashion and written in standard English?

Reviewer #2: No

Reviewer #2: The authors ignored my comment #2, so that the cited literature remains essentially an acknolwedgment to Chinese authors working in the field. Moreover, the reply letter does not indicate any action to address my comment #1.

I thus stopped at comment #2 and concluded that the authors decided to neglect my comments.

I am recommending alternative reviewers to the journal.

**Do you want your identity to be public for this peer review?** For information about this choice, including consent withdrawal, please see our Privacy Policy

Reviewer #2: No

---

## [Author Response · Author response to Decision Letter 2]

23 Jan 2025

1. If the authors have adequately addressed your comments raised in a previous round of review and you feel that this manuscript is now acceptable for publication, you may indicate that here to bypass the “Comments to the Author” section, enter your conflict of interest statement in the “Confidential to Editor” section, and submit your "Accept" recommendation.

Reply : We express our heartfelt thanks to the reviewers for their detailed review work. In this revision, we have added literature from non-Chinese authors to enhance the comprehensiveness of citations.

2. Is the manuscript technically sound, and do the data support the conclusions?

Reviewer #2: Partly

Reply : This paper is based on Fluent software, and the data are obtained by the calculation and post-processing function of Fluent software. The calculation process and control equations strictly follow the Euler two-fluid method, and the calculation results have reached the convergence state, thus ensuring the correctness of the calculation process. In addition, the later data processing is strictly extracted and analyzed according to the calculation file, which ensures the accuracy of data processing.

3. Has the statistical analysis been performed appropriately and rigorously?

Reviewer #2: N/A

Reply : The data statistics in this paper mainly rely on Tecplot post-processing software. The cloud map and wind speed data displayed in the article are imported into the calculation data through the software and processed. The analysis process uses Origin software for scientific drawing and data analysis, and the whole process strictly follows the originality of the calculated data.

4. Have the authors made all data underlying the findings in their manuscript fully available?

Reviewer #2: Yes

Reply : The availability of the data covered in this article has been uploaded to the public repository in accordance with the requirements of your publication. The corresponding URL link is : https://doi.org/10.5061/dryad.76hdr7t5s

5. Is the manuscript presented in an intelligible fashion and written in standard English?

Reviewer #2: No

Reply : This article has undergone rigorous ' spelling proofreading ' and ' grammar proofreading ', and has been carefully polished for the text, aiming to ensure the optimization of the article structure, the correction of grammar and spelling errors, the adjustment of language expression, and the follow-up of academic norms to meet the publishing standards of your journal.

6. Review Comments to the Author

Reviewer #2: The authors ignored my comment #2, so that the cited literature remains essentially an acknolwedgment to Chinese authors working in the field. Moreover, the reply letter does not indicate any action to address my comment #1.

I thus stopped at comment #2 and concluded that the authors decided to neglect my comments.

I am recommending alternative reviewers to the journal.

Reply : We apologize for the omission of your reply in our previous reply. A supplementary comment reply is attached.

The references you mentioned earlier have been added to the introduction of this article. Please refer to the introduction for details.See lines 32-33 on the first page and lines 16-22 on the second page.

7. PLOS authors have the option to publish the peer review history of their article (what does this mean?). If published, this will include your full peer review and any attached files.

Do you want your identity to be public for this peer review? For information about this choice, including consent withdrawal, please see our Privacy Policy.

Reviewer #2: No

Reply : I decided to remain anonymous.

---

## [Editor Report · Decision Letter 2]

7 Mar 2025

Dear Dr. Li,

Thank you for submitting your manuscript to PLOS ONE. After careful consideration, we feel that it has merit but does not fully meet PLOS ONE’s publication criteria as it currently stands. Therefore, we invite you to submit a revised version of the manuscript that addresses the points raised during the review process.

The authors are suggested to make following changes in the paper:

Include relevant papers in the revised manuscript. 

Aerodynamic shape optimization of barriers for windblown sand mitigation using CFD analysis (by Horvat et al. in Journal of Wind Engineering and Industrial Aerodynamics, paper 104058)Wind tunnel experiment and CFD analysis of sand erosion/deposition due to wind around an obstacle (by Tominaga et al. in Journal of Wind Engineering and Industrial Aerodynamics, vol.182, 2018, pp. 262-271)

Grid independence test results should be added in the paperThe boundary conditions are not clear. The surfaces which are symmetric should be clearly shown on Figure and mentioned in the relevant section of the paper.The properties of HDPE are not given in the paper.The contours in Fig. 5 are similar and the effect of fence height is not noticed. Similarly, the contours of sand thickness distribution in Fig. 8 for four cases do not differ. Better Figures are required to explain the effect of fence height.  

We look forward to receiving your revised manuscript.

Kind regards,

Muhammad Shakaib, PhD

Academic Editor

PLOS ONE
---

## [Author Response · Author response to Decision Letter 3]

14 Apr 2025

Dear editor:

First of all, thank you very much for taking the time to read and evaluate my manuscript carefully in your busy work. Your valuable comments and suggestions are of great significance to me. Secondly, in your letter, I noticed that you have put forward some opinions and suggestions on the format of my manuscript. I have further modified the format of the paper to make it conform to the requirements of the format of your paper. At the same time, if there are other problems in the paper, I hope you can tell me that I will modify and reply in time, and hope that the manuscript can be published in your journal as soon as possible. Finally, I wish you and the review experts a happy life and a smooth work ! Thank you again !

1.Include relevant papers in the revised manuscript. 

Reply : Two papers have been added to the article, as shown in line p1 33-39.

2.Grid independence test results should be added in the paper.

Reply:Because the sand-blocking measures such as sand-blocking fences in this paper are network structures, it is necessary to use the porous medium method to characterize the porosity size. The grid division is divided by array, and other division methods cannot accurately simulate porosity. Therefore, the grid independence verification in this paper is only carried out for the subgrade, and three grid division methods are adopted, which are : extracting the horizontal wind speed at a height of 0.1 m on the top of the embankment. From the Figure.4, the grid division method does not affect the wind speed. The horizontal wind speed at different positions under the three division methods almost overlaps, and the wind speed error does not exceed 1 %. Therefore, the grid division method does not affect the wind speed. It is proved that the model grid in this paper is irrelevant.

3.The boundary conditions are not clear.The surfaces which are symmetric should be clearly shown on Figure and mentioned in the relevant section of the paper.

Reply : The coordinates have been re-added and the boundary condition description is supplemented in the figure, as shown in Figure 2.

4.The properties of HDPE are not given in the paper.

Reply : The main component of HDPE raw material is high-density polyethylene, which is processed by high-strength composite material combined with special weaving process. It has strong anti-ultraviolet radiation, frost resistance, decay resistance, acid, alkali, salt corrosion resistance, long service life and other characteristics. The unique scientific design structure of the sand-proof net can not only make the net body shake with the wind, but also make it have self-purification function and avoid dust accumulation, As shown in line p3 12-17.

5.The contours in Fig. 5 are similar and the effect of fence height is not noticed. Similarly, the contours of sand thickness distribution in Fig. 8 for four cases do not differ. Better Figures are required to explain the effect of fence height.  

Reply: The influence of fence height on the structure of the flow field in Figure 5 has been re-added to the magnified picture of the local low-speed vortex zone, as shown in Figure 6, and the analysis content has been added again, such as p8 16-32 lines and p9 1-2 lines.

In Fig.8, the influence of different sand barrier heights on the distribution of sand accumulation is extracted in detail in Table 3.It can be seen from the table that the thickness of sand accumulation increases with the height of sand-blocking fence between the first and second sand-blocking fences and between the second and third sand-blocking fences. When the height of sand-blocking fence is 2.5m, the thickness of sand accumulation is 50.51 % and 58.33 % higher than that of 2m high sand-blocking fence. It has a better blocking effect on sand particles in the airflow, and the effect of sand-blocking is the most significant. When the height increases to 3.5m, the thickness of sand accumulation does not increase significantly.It can be seen that with the increase of the height of the sand-blocking fence, the thickness of the sand between the first and second sand-blocking fences and between the second and third sand-blocking fences increased compared with the height of the sand-blocking fence at the site of 2m. When the height of the sand-blocking fence increases to more than 2.5m, the thickness of the sand reaches a saturated state and no longer increases. It can be seen that the height of the sand-blocking fence should not be too low. Too low will lead to the inability to effectively block the deposition of sand particles and then move forward, which will cause sand burial and other phenomena. The high windproof efficiency of the sand barrier has not been greatly improved, and it will bring the risk of being blown down by the strong wind while increasing the cost. In summary, when the height of the sand barrier is increased to 2.5 m, it is the most reasonable height and has a better wind and sand resistance effect.

---

## [Editor Report · Decision Letter 3]

23 Apr 2025

Dear Dr. Li,

Thank you for submitting your manuscript to PLOS ONE. After careful consideration, we feel that it has merit but does not fully meet PLOS ONE’s publication criteria as it currently stands. Therefore, we invite you to submit a revised version of the manuscript that addresses the points raised during the review process.

**The authors are suggested to make following changes in the paper**
**(i) The title of section 2.2.3 should include terms 'fluid flow' or 'fluid dynamics' such as 'Governing equations for fluid flow ' (instead of Mechanical equation). Similarly 'Control equation', section 2.2.4, should be renamed e.g. Turbulence modeling.  **
**(ii) HDPE should be deleted from the title as its properties (density, tensile strength etc.) are not used in simulations.**

We look forward to receiving your revised manuscript.

Kind regards,

Muhammad Shakaib, PhD

Academic Editor

PLOS ONE
---

## [Author Response · Author response to Decision Letter 4]

28 Apr 2025

(i)The title of section 2.2.3 should include terms 'fluid flow' or 'fluid dynamics' such as 'Governing equations for fluid flow ' (instead of Mechanical equation). Similarly 'Control equation', section 2.2.4, should be renamed e.g. Turbulence modeling.  

Reply : The titles of sections 2.2.3 and 2.2.4 have been changed as required.

(ii) HDPE should be deleted from the title as its properties (density, tensile strength etc.) are not used in simulations.

Reply : HDPE in the title of the article has been deleted.

---

## [Editor Report · Decision Letter 4]

2 May 2025

Evaluation of protection benefit of sand barrier fence with different heights on desert highway

PONE-D-24-35646R4

Dear Dr. Li,

We’re pleased to inform you that your manuscript has been judged scientifically suitable for publication and will be formally accepted for publication once it meets all outstanding technical requirements.

Kind regards,

Muhammad Shakaib, PhD

Academic Editor

PLOS ONE